# Diastolic Dysfunction with Vascular Deficits in HIV-1-Infected Female Humanized Mice Treated with Antiretroviral Drugs

**DOI:** 10.3390/ijms26083801

**Published:** 2025-04-17

**Authors:** Fadhel A. Alomar, Prasanta K. Dash, Mahendran Ramasamy, Zachary L. Venn, Sean R. Bidasee, Chen Zhang, Bryan T. Hackfort, Santhi Gorantla, Keshore R. Bidasee

**Affiliations:** 1Department of Pharmacology, College of Clinical Pharmacy, Imam Abdulrahman Bin Faisal University, Dammam 31441, Saudi Arabia; falomar@iau.edu.sa; 2Departments of Pharmacology and Experimental Neuroscience, College of Medicine, University of Nebraska Medical Center, Omaha, NE 68198-5800, USA; pdash@unmc.edu (P.K.D.); mramasamy@unmc.edu (M.R.); zvenn@unmc.edu (Z.L.V.); sbidasee2@huskers.unl.edu (S.R.B.); chen.zhang@unmc.edu (C.Z.); sgorantla@unmc.edu (S.G.); 3Cellular and Integrative Physiology, Omaha, NE 68198, USA; bryan.hactkort@unmc.edu; 4Environmental, Agricultural and Occupational Health, University of Nebraska Medical Center, Omaha, NE 68198-5800, USA; 5Nebraska Redox Biology Center, Lincoln, NE 68503, USA; 6Center for Heart and Vascular Research, University of Nebraska Medical Center, Omaha, NE 68198, USA

**Keywords:** heart failure with preserved ejection fraction (HFpEF), diastolic dysfunction, HIV-1 infection, humanized mice, antiretroviral drugs

## Abstract

Early-onset heart failure is a major treat to healthy aging individuals with HIV-1 infection. Women with HIV-1 infection (WLWH) are especially vulnerable and develop heart failure with preserved ejection fraction (HFpEF), of which left ventricular diastolic dysfunction, vascular deficits, myocardial infarction, and fibrosis are major components. HIV-infected rodent models that exhibit these pathophysiological features remain under-reported, and this has left a void in our understanding of their molecular causes and therapeutic strategies to blunt its development. Here, we show that female NOD.Cg-Prkdc^scid^Il2rgt^m1Wjl^/SzJ humanized mice (Hu-mice) infected with HIV-1_ADA_ and treated for 13 weeks with dolutegravir (DTG)/tenofovir disoproxil fumarate (TDF)/emtricitabine (FTC) develop progressive diastolic dysfunction with preserved ejection fraction (E:A ratio, E:e′, IVRT, left atrial volume and global longitudinal strain increased by 32.1 ± 5.1%, 28.2 ± 5.6%, 100.2 ± 12.6%, 26.6 ± 4.2% and 32.5 ± 4.3%, respectively). In vivo photoacoustic imaging revealed a 30.4 ± 6.8% reduction in saturated oxygenated hemoglobin in the anterior wall of the heart. The ex vivo analysis of hearts showed a reduction in density of perfused microvessels/ischemia (30.6 ± 6.2%) with fibrosis (20.2 ± 1.2%). The HIF-1α level was increased 2.6 ± 0.5-fold, while inflammation-induced serum semicarbazide amine oxidase and glycolysis byproduct methylglyoxal increased 2-fold and 2.1-fold, respectively. Treating H9C2 cardiac myocytes with DTG, FTC and TDF dose-dependently increased expression of HIF-1α. These data show that HIV-infected Hu-mice treated with DTG/TDF/FTC for thirteen weeks develop cardiac diastolic dysfunction, with vascular deficits, ischemia, and fibrosis like those reported in women living with HIV-1 infection (WLWH). They also show that DTG, TDF, and FTC treatment can increase total HIF-1α in H9C2 cells.

## 1. Introduction

Highly active combinations of antiretroviral drugs (ARDs) have increased life expectancy for the majority of the ~40 million individuals living with HIV-1 infection (PLWH) [1,2]. However, new challenges are emerging as PLWH are developing early onset heart failure [3,4,5,6,7,8]. Women living with HIV infection (WLWH) are especially vulnerable as they have a two–three-fold higher risk of developing heart failure with preserved ejection fraction (HFpEF), of which diastolic dysfunction (DD), vascular deficits, myocardial infarction (ischemia), and fibrosis are major components. The prevalence of DD ranges from 19 to 47%, with onset between 40 and 50 years of age [7,9,10]. WLWH that have HFpEF have longer hospitalization stays with higher mortality rates than men with HIV-1 infection and HFpEF [11]. To date, animal models that are infected with HIV and treated with ARDs that exhibit diastolic dysfunction (DD), vascular deficits, myocardial infarction (ischemia), and fibrosis remain under-reported, and this has resulted in a paucity of information for the underlying causes of early-onset cardiac DD in WLWH and pharmacological strategies to blunt its development.

Cardiac DD has been previously reported in HIV transgenic rodent models and SIV-infected rhesus macaque monkeys. Mak et al. found both diastolic and systolic dysfunctions in male HIV-1 transgenic rats (contains HIV-1 provirus with a functional deletion of *gag* and *pol*), and these changes were not attenuated with ARD treatment. These workers also reported that ARDs (tenofovir/emtricitabine + atazanavir/ritonavir) were triggering oxidative/nitrosative stress, hypomagnesemia, and substance P-dependent neurogenic inflammation in HIV-1 transgenic rats, and magnesium supplementation blunted diastolic and systolic dysfunctions [12]. Using HIV-1 transgenic mice (Tg26 that overexpress HIV proteins), Chueng et al., also found a reduction left ventricular diastolic volume with no change in systolic function (ejection fraction and fractional shortening), consistent with heart failure with preserved ejection fraction (HFpEF) [13]. They also reported that the increase in left ventricular stiffness was due in part to the downregulation of Bcl2-associated athanogene 3 (BAG3) [13]. The sexes of mice used in the latter study were not defined. Since levels of HIV proteins are typically low in PLWH on antiretroviral drugs, HIV transgenic rodents do not reflect the condition in PLWH.

Others have also shown that rhesus macaque monkeys infected with SIV develop cardiac DD [14]. The DD in SIV-infected rhesus macaques correlated with the extent of SIV replication in the myocardium, suggesting that the replication of SIV-1 in myocardial macrophages is inducing cardiomyocyte damage and DD [14]. In another study, these workers also showed that treatment with the CCR5 inhibitor maraviroc attenuated the cardiac DD, indicating that SIV infection was the underlying cause for the development of DD. The latter is consistent with the notion that SIV/HIV are risk enhancers for heart failure [15,16]. Burdo and colleagues also reported that osteopontin is an integral mediator of cardiac interstitial fibrosis in SIV infected macaque monkeys. However, no cardiac function data were given [17]. Monkeys are expensive to maintain, limiting this model to only a few laboratories.

Earlier, we reported that humanized NOD.Cg-Prkdc^scid^IL2rγ^tm1Wjl^/SzJ (NSG) mice infected with HIV-1 developed progressive diastolic and systolic dysfunctions. The onset of heart failure was also earlier in female mice [18]. This mouse model has the advantage over HIV transgenic models, in that they can be infected with HIV-1 and plasma HIV-1 viremia can be lowered by treating with ARDs. What is not yet clear is whether female NSG-humanized mice infected with HIV-1 and treated with antiretroviral drugs also develop DD with vascular deficits, ischemia, and fibrosis.

More than 30 antiretroviral drugs in nine mechanistic classes are approved by the U.S. Food and Drug Administration (FDA) for treating HIV infection in PLWH. These nine classes include nucleos(t)ide reverse transcriptase inhibitors (NRTIs), non-nucleoside reverse transcriptase inhibitors (NNRTIs), protease inhibitors (PIs), integrase strand transfer inhibitors (INSTIs), a fusion inhibitor, a CCR5 antagonist, a CD4 T lymphocyte (CD4) cell post-attachment inhibitor, a gp120 attachment inhibitor, and a capsid inhibitor. In addition, two drugs, ritonavir (RTV, a PI) and cobicistat (COBI, CYP3A inhibitor), are used as pharmacokinetic (PK) enhancers (or boosters) to improve the PK profiles of PIs and the INSTI elvitegravir (EVG) [19]. Improvements in potency, tolerability, safety, convenience, drug–drug interactions, and genetic barriers to the emergence of drug resistance have led to recommendations for antiretroviral drug combinations for people with HIV-1 infection. Two of these first line recommendations are dolutegravir (DTG)/tenofovir alafenamide (TAF) or tenofovir disoproxil fumarate (TDF)/emtricitabine (FTC) or lamivudine (3TC) [19]. In the ADVANCE Trial, Hill and colleagues indicated that in spite of similar HIV suppression, PLWH on TAF/FTC/DTG regimen experienced greater weight gain and clinical obesity than PLWH on TDF/FTC/DTG after week 192, and that the weight gain was particularly pronounced in WLWH [20]. Thus, the purpose of the present study was to use longitudinal echocardiography, photoacoustic imaging, biochemical, histochemical, fluorescence, and Western blot assays to determine if female NSG mice infected with HIV-1 and treated with DTG/TDF/FTC develop cardiac DD with vascular deficits, ischemia, and fibrosis. We also investigated whether DTG, TDF, and FTC can induce the expression of hypoxia-inducible factor, HIF-1α in H9C2 cardiac myocytes. Some of the data were presented at CROI, 2024 [21].

## 2. Results

### 2.1. Characterization of Humanized Mice Used in Study

At the start of the study, human CD45+ immune cell reconstitution in the eighteen female Hu-NSG mice was used for this study and ranged from 15 to 40% of peripheral blood white blood cells. HIV-1 infection led to productive infection, with plasma HIV-1 viral loads peaking at eight weeks post-infection (wpi, 1.8 ± 0.2 × 10^5^ RNA copies/mL), and maintaining after 17 weeks post-infection (1.6 ± 0.2 × 10^5^ RNA copies/mL at 17 weeks (Table 1)). After one week of treatment with DTG/TDF/FTC, the plasma viral load of HIV-1-infected mice decreased to 1.0 ± 0.2 × 10^3^ RNA copies/mL, and after 12 weeks of treatment the plasma viral load decreased to 50 copies/mL (Table 1). The percentages of CD4+ T cells in the blood declined in HIV-1-infected Hu-NSG mice from 72.6 ± 0.4 to 37.2% ± 2.1, while CD8+ T cells gradually increased over the same period from 29.2 ± 1.3 to 50.2% ± 6.1 (Figure 1). All animals maintained their body masses to within 5% of their starting weights (Table 1). When hearts were excised after euthanasia and weighed, the mean heart weight of HIV-1-infected Hu-NSG mice was significantly lower than that of aged-matched uninfected controls (Table 1). Treating HIV-1-infected Hu-mice with DTG/TDF/FTC for 12 weeks did not blunt the decrease in heart weight (Table 1).

### 2.2. Echocardiographic Analyses

Echocardiography was used to assess blood flow velocities, dimensions, and contractile kinetics of the left chambers of the heart. Speckle tracking (ST) was also used to determine segmental tissue motion of the heart in multiple planes over the cardiac cycle to provide information on myocardial strain.

(1) Diastolic function: Figure 2A shows representative pulsed wave doppler images (mitral valve) generated in hearts from uninfected controls, HIV-1-infected, and HIV-1-infected mice treated with DTG/TDF/FTC at the end of the in vivo protocol. Three out of seven female HIV-1-infected Hu-mice exhibited mitral regurgitation after seventeen weeks of infection, as indicated by the present of an L-wave (see yellow arrow). All female mice infected with HIV-1 developed grade III diastolic dysfunction after 17 weeks of infection, indicated by an increase in E (peak early transmitral velocity):A (peak late transmitral velocity) ratio, Figure 2B, red line. The increase in the E:A ratio in HIV-1-infected mice was also progressive, reaching a peak of 3.1 ± 0.2 after seventeen weeks of infection (Figure 2B red line). After five weeks of infection, one mouse exhibited mild diastolic dysfunction with an E:A ratio of 0.89. This same mouse developed grade III DD after 17 weeks of infection. Similar increases were seen in the E:e′ ratio and isovolumetric relaxation time (IVRT, Figure 2D,E). MV deceleration also progressively decreased in HIV-infected humanized mice. E:A and E:e′ ratios in HIV-1-infected Hu-mice treated with DTG/TDF/FTC were lower than that of HIV-1-infected mice, but differences were not significant from HIV-1-infected animals (*p* > 0.05) (Figure 2B,C and green line). DTG/TDF/FTC treatment did not blunt changes in IVRT and MV deceleration times (Figure 2D,E green lines).

(2) Systolic function: After five weeks of HIV-1 infection, % ejection fraction (% EF) significantly decreased in female mice (*p* < 0.05), Figure 3A,B. However, there was no change in % fractional shortening (% FS) (Figure 3B). Both % EF and % FS progressively worsen as the duration of infection increased (Figure 3B,C). Treating female HIV-1-infected female humanized mice with DTG/TDF/FTC for 13 weeks progressively blunted decreases in % EF and % FS (Figure 3B,C).

(3) Left atrial volume*:* Left atrial volume was also measured in uninfected, HIV-1-infected, and HIV-1-infected female humanized mice treated with DTG/TDF/FTC for 13 weeks. Figure 4B shows that, after nine weeks of infection, the left atrial volume of was 2.2-fold higher than that of uninfected animals (red line), and the increase in the left atrial volume persisted through 17 weeks of infection. Figure 4A (middle panel) also shows a representative M-Mode recording of increased left atria size in HIV-1-infected female mice. Treating HIV-1-infected animals with DTG/TDF/FTC for 13 weeks did not blunt the increase in left atrial volume (Figure 4A,B, green line).

(4) Speckle tracking (ST): ST analyses of B-mode images using the “normal peak algorithm” (systole) did not show any significant differences in the global longitudinal strain after five weeks of infection (Figure 5A, red line). However, as the duration of infection increased, the global longitudinal strain increased, nearly doubling that of uninfected controls after 17 weeks of infection. Figure 5B shows the representative long axis velocities during three to six consecutive cardiac cycles (left), and radial (upper) and longitudinal (lower) strains in six opposite segments (right) from an uninfected, HIV-infected, and HIV-1-infected female mice treated with DTG/TDF/FTC for seventeen weeks. All seven mice in the HIV-1-infected group exhibited dyskinesia (expansion of a wall segment during systole) and this was blunted with DTG/TDF/FTC treatment (red arrows).

(5) Photoacoustic imaging: Photoacoustic imaging (PAI) was performed in uninfected, HIV-1-infected, and HIV-1-infected animals treated with DTG/TDF/FTC to assess the co-registration of the photoacoustic signal from hemoglobin with an anatomic view of the heart in real-time during diastole and systole to enable the calculation of mean saturated oxygen concentration in mice following the inhalation of room air and 100% oxygen. Figure 6A(i) shows a saturated hemoglobin-oxygen signal in the anterior wall of the heart during diastole when uninfected anesthetized animals were administered room air (yellow arrow). The region of interest was drawn to capture the entire anterior/septal wall in the plane of image taken. The intensity of the saturated hemoglobin-oxygen signal in the region of interest increased when anesthetized uninfected animals were administered 100% oxygen (Figure 6A(i) right panel and Figure 6B). When HIV-infected anesthetized animals were administered room air, the intensity of the saturated hemoglobin-oxygen signal in the region of interest was 14.3 ± 1.2% less than that of uninfected animals (Figure 6A(ii) left panel and Figure 6B). Exposing HIV-infected anesthetized animals to 100% oxygen increased the intensity of the saturated hemoglobin-oxygen, but the amount was 22.2 ± 2.2 % less than that of uninfected animals (Figure 6A(ii) right panel and Figure 6B). Exposing anesthetized HIV-infected animals treated with DTG/TDF/FTC to room air also generated a less intense saturated hemoglobin-oxygen signal in the region of interest compared with that of uninfected animals (Figure 6A(iii) left panel and Figure 5B). The intensity of the saturated hemoglobin-oxygen signal obtained when DTG/TDF/FTC-treated HIV-infected anesthetized animals were exposed to 100% oxygen was similar to that of HIV-infected anesthetized animals.

### 2.3. Histopathological Analyses

(i) Microvascular integrity*:* BSA-FITC was injected via tails vein into uninfected, HIV-1-infected, and HIV-1-infected mice treated with DTG/TDF/FTC five minutes prior to euthanasia to assess the integrity of the blood microvasculature in their hearts. In left ventricular tissues from uninfected mice, the green fluorescence of BSA-FITC was seen throughout the myocardium and contained within blood vessels (Figure 7A(i), yellow arrows), indicative of a high density of perfused microvessels. In HIV-1-infected mice, the BSA-FITC fluorescence was heterogeneously distributed, with some regions showing green BSA-FITC fluorescence and regions without (Figure 7A(ii), white arrow). The reduction in the density of perfused microvessels was 48.1 ± 8.2 %. Treating HIV-infected animals with DTG/TDF/FTC for thirteen weeks blunted the reduction in the density of perfused microvessels Figure 7A(iii). The density of perfused microvessels was 79.4 ± 10.2% that of uninfected controls. The density of microvessels perfused with BSA-FITC per 20× frame from hearts from uninfected, HIV-1-infected, and HIV-1-infected mice treated with DTG/TDF/FTC are shown in Figure 7B.

(ii) Fibrosis*:* Photomicrographs of Masson’s Trichrome-stained hearts from uninfected controls exhibited minimal interstitial and perivascular fibrosis (Figure 8A(i). However, photomicrographs of Masson’s Trichrome-stained hearts from HIV-1-infected mice exhibited significantly more interstitial and perivascular blue staining, suggestive of fibrosis. Treating HIV-1-infected mice with DTG/TDF/FTC for thirteen weeks was attenuated. Mean data are presented in Figure 8B.

(iii) Western blot analyses: Since the hearts of HIV-infected and HIV-1-infected mice treated with DTG/TDF/FTC showed reductions in the density of perfused microvessels (i.e., ischemia), Western blots were conducted for ischemia-induced protein, HIF-1α. In this study, cardiac HIF-1α increased two-fold after 17 weeks of infection (Figure 9) and this increase in HIF-1α was blunted in HIV-1-infected animals treated with DTG/TDF/FTC for thirteen weeks.

Prior studies have shown that dolutegravir, tenofovir disoproxil fumarate, and emtricitabine are potent inducers of reactive oxygen species (ROS) [22,23]. In cells, HIF-1α is continuously synthesized and transported to the cytoplasm of cells where it gets hydroxylated by prolyl hydroxylase and is a target for proteasomal degradation [24]. Since ROS is a potent inhibitor of prolyl hydroxylase, we investigate whether the increase in HIF-1α seen in cardiac tissues from HIV-1-infected animals treated with DTG/TDF/FTC was due in part to ARD drugs. For this, H9C2 cells were treated with DTG, TDF, or FTC for 2 h, and Western blots were conducted. In this study, 7.8 3M of DTG, TDF, and FTC induced robust increases in expression of total HIF-1α (>3-fold) in H9C2 cells. (Figure 10). Higher concentrations of TDF also dose-dependently increased the expression of HIF-1α, while higher concentrations of [DTG] and [FTC] did not increase the expression of HIF-1α further (Figure 10).

## 3. Discussion

Early-onset heart failure continues to be a major threat to healthy aging in people with HIV-1 infection. Women living with HIV-1 infection (WLWH) are especially vulnerable to developing HFpEF of which DD, vascular deficits, ischemia, and fibrosis are major components [11,25,26,27,28]. To date, it is not clear whether humanized mouse models infected with HIV-1 and treated with ARDs also develop DD, vascular deficits, ischemia, and fibrosis.

Here we show for the first time that female HIV-1-infected NSG humanized mice treated with DTG/TDF/FTC develop a progressive DD, with no change in % EF and % FS, akin to that seen in WLHW [21,28,29]. The DD in HIV-infected NSG humanized mice treated with ARDs manifest as increases in the E:A ratio arising from a reduction in the A-wave velocity, E wave deceleration time, IVRT, global longitudinal strain, and left atria volume. A reduction in A-wave amplitude (velocity), with no change or increase in E-wave amplitude, is consistent with Grade III DD (restrictive filling) [30]. A shortened E wave deceleration time is indicative of reduced left ventricular compliance and/or increased in left atrial pressure [31]. In this study, global longitudinal strain was increased in HIV-1-infected female humanized mice treated with DTG/TDF/FTC, consistent with increased stiffness [32]. An increase in IVRT is indicative of impaired ventricular relaxation, which can lead to decreased cardiac filling, increased left atrial pressure, and, ultimately, a higher risk of developing DD. DTG/TDF/FTC-treated female HIV-1-infected mice also exhibited an enlarged left atria volume, suggestive of increased left atrial pressure [33,34]. The heart weight:body weight ratios in DTG/TDF/FTC-treated HIV-1-infected mice were not significantly different from those of uninfected animals (122.8 ± 8.1 vs. 123.8 ± 5.2%, respectively, also see Table 1). Four of the five DTG/TDF/FTC-treated mice also exhibited dyskinesia (expansion of a wall segment during systole), although the extent was significantly lower than that seen in HIV-1-infected mice [18]. In future studies invasive pressure-volume loops will be assessed to validate DD.

In this study we found that hearts from HIV-1-infected female mice with and without DTG/TDF/FTC treatments exhibited heterogeneity in the density of perfused microvessels, with some regions exhibiting a high density of perfused microvessels, while other regions contained a low density of perfused microvessels (Figure 7). Since blood vessels transport oxygen/nutrients to and remove waste from myocytes, these data suggest that some myocytes are residing in “hypoxic and low nutrient environments” while other myocytes are in regions with ample oxygen and nutrients. Impaired vascular perfusion/ischemia has been reported WLWH [26,28]. This heterogeneity in oxygen/nutrients will have significantly functional implications. First, low oxygen environment will result in sarcolemmal disruption and impairment in myofibril relaxation [35]. Second, myocytes low oxygen regions will shift from oxidative phosphorylation to glycolysis and fatty acid oxidation to generate the ATP needed for functioning [36,37]. The shift from oxidative phosphorylation to glycolysis will also result in an increase in expression of HIF-1α, to support expression of glycolysis-related genes [24]. Since perfusion is compromised, substrates needed for glycolysis and fatty acid oxidation will be limiting, and eventually myocytes will die follow prolonged ischemia. Third, due to the negligible regeneration capacity of the myocardium, ischemic regions will heal by scar formation [38]. Fourth, scarred regions in the left ventricle will have altered electrical conduction and will result in non-synchronous contractions in ischemic and the normal regions [39]. The latter may help explain the dyskinesia seen in HIV-1-infected female humanized mice treated with DTG/TDF/FTC. Whether “hypoxic myocytes” respond differently to intrinsic ligands and antiretroviral drugs compared to “normoxic myocytes” remains unresolved and more work is needed to address this. The scar can also serve as an arrhythmogenic substrate [39] and could help explain the increased incidence of sudden cardiac death in PLWH.

Dolutegravir, tenofovir disoproxil fumarate, and emtricitabine are potent inducers of reactive oxygen species (ROS) [22,23]. Since ROS inhibits prolyl hydroxylase, the enzyme that hydroxylate HIF-1α and target it for degradation [24], we investigate whether the increase in HIF-1α seen in cardiac tissue from HIV-1-infected animals treated with DTG/TDF/FTC was due in part to ARD drugs. In this study we found that exposing H9C2 cardiac myocytes to DTG, TDF, FTC for 2 h. increased expression of HIF-1α nearly three-fold These data indicate that DTG, TDF, FTC may be contributing to the increase in expression of HIF-1α seen in cardiac tissues from HIV-infected NSG mice treated with ARDs. Whether the diastolic dysfunction is linked to the elevation in HIF-1α remains to be determined.

In this study, we also found that hearts from female HIV-1-infected mice with and without DTG/TDF/FTC treatment contained significant amounts of perivascular and interstitial fibrosis as indicated by the blue Trichrome Masson staining. An increase in cardiac fibrosis will increase global longitudinal strain (stiffness). Although, the mechanism for the increased myocardial fibrosis is not clear, transcytosis of substances including immune cells from the blood into the myocardium will trigger inflammation [40] activate matrix metalloproteinases (MMP’s) and increase deposition of collagen fibers [41]. An increase in myocardial inflammation and the glycolysis metabolite methylglyoxal will also activate the inflammation-induced transcription factor NF-κB (p-P65) and the inflammasomes to increase expression of an array of inflammatory mediators [42,43]. Additional work will be needed to define mechanisms responsible for interstitial and perivascular fibrosis.

In conclusion, the present study shows for the first time that HIV-1-infected female Hu- mice treated with DTG/TDF/FTC develop a progressive DD, with vascular deficits, myocardial ischemia, and fibrosis. This model should allow us to delineate mechanisms that contribute to cardiac diastolic dysfunction in HIV-1-infected female Hu- mice treated with DTG/TDF/FTC and identify druggable targets and treatment strategies to attenuate it.

### Limitations

This study is not without limitations. Although the development, filling, and emptying of a mouse heart are similar to those of humans, subtle differences exist that prevent mice from fully replicating the cardiac pathophysiologies seen in WLWH [44]. One prominent difference is that mouse hearts beat 500× per minute and human hearts beat ~80× per minute, but the mechanisms that drive automaticity, contraction, and relaxation differ only slightly. Our group uses NOD.Cg-Prkdc^scid^Il2rgt^m1Wjl^/SzJ (NSG) to generate humanized NSG mice for our studies. NSG humanized mice (i) do not have fully developed B cells and lymph nodes, (ii) metabolize drugs differerently from humans due to differences in expression of liver enzyme, and (iii) have a shorter lifespan than that of non-humanized mice [45]. However, the cardiac contraction and relaxation kinetics of NSG humanized mice are similar to those of NSG non-humanized mice in terms of E:A ratio (~1.2–1.3), ejection fraction (65–70%), and fractional shortening (35%). After humanization, <5% of animals develop graft vs. host diseases, but GVH mice are not used for experiments. To ensure that stress is not a response to the changes in cardiac function, the three groups of NSG humanized mice used for this study were housed under identical conditions and given the same feed and enrichment environment. To minimize handling stress, after transportation to the echocardiographic facilities, all animals were left for at least one hour to acclimatize before cardiac function was measured.

## 4. Material and Methods

### 4.1. Ethics Statement

Animals used in this study were approved by the University of Nebraska Medical Center (UNMC) Institutional Animal Care and Use Committee (IACUC) #21-100-06 FC, reapproved on 02/12/2025 (heart and pulmonary deficits in HIV-1 infection and IBC #23-07—17 (End Organ deficits in HIV-1 infection and Lupus Nephritis)), reapproved 28 July 2023. All animal work were performed in compliance with UNMC institutional policies and NIH Guide for Use and Care of Laboratory.

### 4.2. Antibodies and Reagents

Magnetic beads conjugated CD34+ antibodies for isolation of human hematopoietic stem cells from cord blood were from Miltenyi Biotec Inc., Auburn, CA, USA. Primary antibodies against CD45, CD19, CD14, CD8, CD4, CD3 and CD68 were from BD Pharmingen, San Diego, CA, USA). Rabbit mAb anti HIF-1α antibodies were obtained from Cell Signaling (D2U3T, Danvers, MA, USA), β-actin antibody was from ThermoFisher Scientific (Watham, MA, USA), and donkey anti rabbit IgG-HRP, and donkey anti rabbit and goat IgG-HRP were from Santa Cruz Biotechnology Inc. (Santa Cruz, CA, USA). Fluorescein isothiocyanate-labelled bovine serum albumin (BSA-FITC), and Trichrome Masson staining kits were from Sigma-Aldrich, St Louis MO. was from Sigma-Aldrich (St Louis, MO, USA). OxiSelect™ Methylglyoxal Competitive ELISA, was from Cell Biolabs Inc. (San Diego, CA, USA). Fluoro-SSAO™ assay kit was from Cell Technology (Mountain View, CA, USA). Pierce™ RIPA buffer was from VWR Scientific (Radnor, PA, USA). Bradford Protein Assay kit was from ThermoFisher Scientific (Waltham, MA, USA). All other reagents were from commercial sources. All other reagents used were of the highest grade commercially available.

### 4.3. Cell Culture

The H9C2 cardiac myocyte cell line was from ATCC (Manassas, VA, USA). Cells were grown in Dulbecco’s Modified Eagle’s Medium (DMEM) containing 4 mM L-glutamine, 4500 mg/L glucose, 1 mM sodium pyruvate, and 1500 mg/L sodium bicarbonate at 37 °C. All experiments were conducted with cells 70–80% confluent.

### 4.4. Generation of Hu-NSG (Humanized) Mice

Humanized mice (Hu-mice) were prepared as described in prior publications [18,46]. The percentage of CD45 cells in humanized mice used for this study ranged from 25 to 50%. About 5% of our Hu mice develop graft vs. host disease (GVHD), and these mice were excluded from our study.

### 4.5. HIV-1 Infection

Twelve female humanized mice were infected intraperitoneally (IP) with 2 × 10^4^ tissue culture infectious dose 50 (TCID_50_) of HIV-1_ADA_ at 20 weeks age. Six uninfected Hu-NSG mice served as aged-matched controls. Peripheral blood samples were collected every four weeks via submandibular vein bleeding to assess HIV-1 viral RNA. Plasma HIV-1 RNA levels were measured using an automated COBAS Ampliprep V2.0/Taqman-48 system (Roche Molecular Diagnostics, Basel, Switzerland) as per the manufacturer’s instructions.

### 4.6. Antiretroviral Drug Treatment

Four weeks post-infection, HIV-1-infected animals were divided into two groups. Group 1 animals were fed a normal chow (protein 25%, carbohydrate 58%, fat 17%), while Group 2 animals were fed normal chow containing DTG, TDF and FTC prepared by Teklab, Research Diets, New Brunswick, NJ to deliver doses of 15 mg/kg/day DTG, 131 mg/kg/day TDF and 87.5 mg/kg/day based on 4 g/day consumption for thirteen weeks. Uninfected control animals were fed normal chow without antiretroviral drugs.

### 4.7. Echocardiography

Transthoracic echocardiography was performed using a Fujifilm Visual Sonics Vevo 3100 system with a LAZR-X (Fujifilm Visual Sonics, Toronto, ON, Canada), employing a MX550D transducer with a center frequency of 40 Hz and an axial resolution of 40 µM, prior to and 5, 9, 13, and 17 weeks after infection with HIV-1 or saline injection for controls. For echocardiographic recordings, mice were anesthetized with 1–3% isoflurane (Cardinal Health, Dublin, OH, USA) and taped in the supine position on a heated 37 °C pad with feet leads. Anesthesia was maintained via a nose cone and pulsed-wave Doppler, M-mode and tissue Doppler images were acquired in the parasternal short axis mode and digitally stored in cine loops. The offline Program Vevo LAB 5.5.1 was then used to assess peak early- and late-diastolic transmitral velocities (E and A waves), E-wave deceleration time, and isovolumetric relaxation time (IVRT) as indices of diastolic function/dysfunction and fractional shortening (% FS), and percent ejection fraction (% EF) were used as measures of systolic functions. E/A ratio and E/e′ ratio were calculated.

### 4.8. Speckle Tracking

Parasternal long axis B-mode images were obtained at a rate of >300 frames/second and digitally stored in cine loops. Vevo LAB 3.2.6 was used to determine the global longitudinal, circumferential strain, and to assess for dyskinesia during systole using six-segment (anterior base, AB; anterior middle, AM; anterior apex, AP, posterior base, PB; posterior middle, PM; and posterior apex PA) analyses.

### 4.9. Photoacoustic Imaging

Photoacoustic imaging (PAI) was performed thirteen weeks post-infection with HIV-1 or saline to assess saturated O_2_ (sO_2_) concentration using the photoacoustic signal from hemoglobin. Imaging was conducted using a Vevo 3100 LAZR-X with the MX550D transducer that operates at a center frequency of 40 MHz and axial resolution of 75 µm; the optical component transmits tunable laser wavelengths from 680 to 970 nm, with a peak pulse energy output of 20 mJ/cm^2^ (FUJIFILM VisualSonics Corporation, Bothell, WA, USA). Mice with chests shaved were anesthetized with 1.5% isoflurane with balance of medical air (21% O_2_) and placed on a heated platform for PAI. Heart rate, determined by ECG electrodes connected to their feet, was kept around 400 beats per minute in all mice. Vevo Lab 5.7.1 (FUJIFILM VisualSonics Corporation, Bothell, WA, USA) software allow for co-registration of the PA signal from Hb with an anatomic view of the heart in real-time during diastole and systole and enable calculation of mean sO_2_ concentration. While recording live imaging, the balance gas for isoflurane was switched to 100% O_2_ to measure real-time changes in myocardial sO_2_. Regions of interest were drawn to capture the entire anterior/septal wall in the plane of image taken.

### 4.10. Histopathological Analyses

(1) Density of perfused microvessels and ischemia: Two days after PAI, mice were injected with BSA-FITC (60 mg/kg in sterile 1× PBS buffer, 50 µL) via a tail vein, as described earlier [18]. Hearts were then collected, fixed in paraformaldehyde, embedded in paraffin, and 5 μm-thick sections were mounted onto pre-cleaned glass slides to assess the density of perfused microvessels. Images were taken using a Nikon TE2000 microscope attached to a Coolsnap HQ2 CCD camera (Photometrics, Tucson, AZ, USA).

(2) Fibrosis: Five micrometer sections from hearts from the three groups of animals embedded in paraffin were de-paraffinized and rehydrated, and fibrosis was assessed using the Masson Trichrome staining kit as per the manufacturer’s instructions. A Zeiss ApoTome inverted microscope was used to assess fibrotic staining with a 10× lens.

(3) Western blot analyses: Frozen hearts were cut into small pieces and added to RIPA buffer (Pierce™ RIPA Buffer, ThermoFisher Scientific, USA) and sonicated for 3 × 10 s. After this, samples were centrifuged for 10 min at 3500× *g* and the protein concentration in each supernatant was determined using the Bradford protein assay kit. Samples (25 µg) were electrophoresed using 4–15% gradient gels for 2 h. at 150 V. Proteins were transferred to PVDF membranes, and Western blots were conducted for HIF-1α with β-actin as reference. Primary antibody concentrations were 1:1000 for 16 h at 4 °C and secondary antibody concentrations were 1:2000 for 2 h at room temperature.

(4) Methylglyoxal and semicarbazide-sensitive amine oxidase in sera: MG-H1 levels (as surrogate for MG) were measured in plasma using OxiSelect™ Methylglyoxal Competitive ELISA, Cell Biolabs Inc., San Diego, CA, USA) as per the manufacturers’ instruction inside a BSL2+ Facility at UNMC. The activity of the non-selective inflammation marker, semicarbazide-sensitive amine oxidase (the soluble form of VAP-1) was also measured in plasma using Fluoro-SSAO™ (Cell Technology, Mountain View, CA, USA) as per the manufacturer’s instruction.

### 4.11. Treatment of H9C2 Cells with Antiretroviral Drugs

H9c2 cardiac myocytes wee grown at 70–80% confluency, and H9c2 cells were treated with varying concentrations of DTG, TDF, or FTC (0, 7.8 µM, 15.6 µM, 31.5 µM, and 62.5 µM) for a 2-h incubation at 37.0 °C. After treatment, cells were washed with ice-cold phosphate-buffered saline (PBS), resuspended in a RIPA buffer (Pierce™ RIPA Buffer, ThermoFisher Scientific, USA) and incubated for 30 min on ice. Samples were then centrifuged at 3500× *g* for 10 min at 4 °C. The pellets were discarded, the supernatants were collected, and protein concentrations were determined using the Bradford method. Samples (25 µg) were electrophoresed on 4–15% gradient gels for 3 h. at 150 V. Proteins were transferred to PVDF membranes and Western blots were conducted for HIF-1α using β-actin. Primary antibody concentrations were 1:1000 for 16 h at 4 °C and secondary antibody concentrations were 1:2000 for 2 h. at room temperature.

### 4.12. Statistical Analysis

Data were analyzed using the GraphPad Prism 7.0 software (La Jolla, CA, USA) and were presented in the text as the mean ± the standard error of the mean. All experiments listed in this manuscript were performed using a minimum of three biologically distinct replicates. The one-way ANOVA with Bonferroni correction for multiple comparisons was used. For studies with multiple time points, two-way factorial ANOVA and Bonferroni’s post-hoc tests for multiple comparisons were performed. Studies used six animals per group. Significant differences were determined at *p* < 0.05.

## Figures and Tables

**Figure 1 ijms-26-03801-f001:**
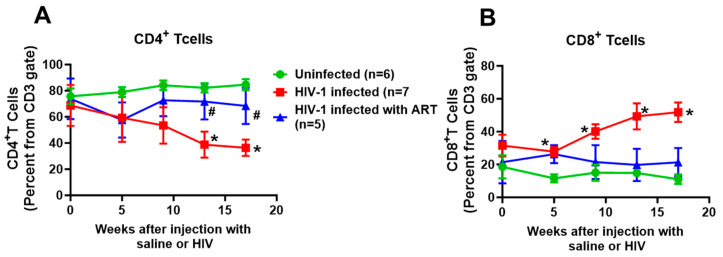
(**A**,**B**) show longitudinal changes in CD4^+^ and CD8^+^T cells in uninfected (n = 6), HIV-1-infected (n = 7), and HIV-infected female humanized mice infected treated with a combination of dolutegravir (DTG 15 mg/kg), tenofovir disoproxil fumarate (TDF, 131 mg/kg,) and emtricitabine (FTC, 87.5 mg/kg) via feed based on 4 g/day consumption for thirteen weeks, starting four weeks after productive infection (n = 5) as a percentage of CD3 gate. All measurements were collected using the LSR-II FACS analyzer (BD Biosciences, Mountain View, CA, USA). * denotes significantly different from uninfected, *p* < 0.05, # denotes significantly different from HIV-1-infected, *p* < 0.05.

**Figure 2 ijms-26-03801-f002:**
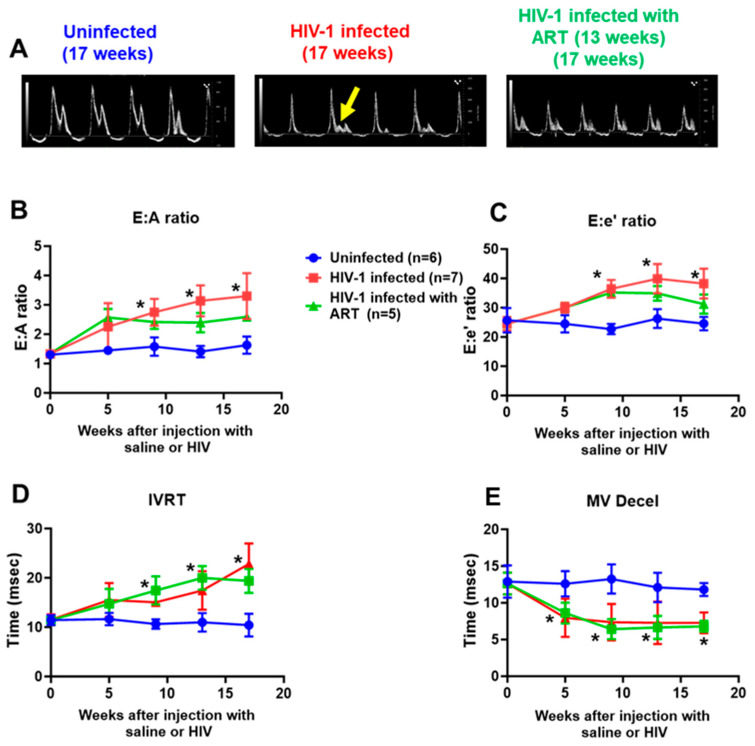
(**A**) shows represented pulsed-wave echocardiograms from uninfected (n = 6), HIV-1-infected (n = 7), and HIV-infected female humanized mice infected with a combination dolutegravir (DTG 15 mg/kg), tenofovir disoproxil fumarate (TDF, 131 mg/kg,) and emtricitabine (FTC, 87.5 mg/kg) via feed based on 4 g/day consumption for thirteen weeks, starting four weeks after productive infection (n = 5). Yellow arrow shows the presence of a L-wave, indicative of mitral regurgitation. (**B**–**E**) show longitudinal changes in E:A and E:e′ ratios, IVRT and MV deceleration time for uninfected, HIV-1-infected, and HIV-infected female humanized mice infected treated with DTG/TDF/FTC for thirteen weeks, starting four weeks after productive infection (n ≥ 5 mice per group). All measurements were taken using Fujifilm Visual Sonics Vevo 3100 system (Fujifilm Visual Sonics, Toronto, ON, Canada) employing a MS550D transducer with a center frequency of 40 Hz and an axial resolution of 40 µM. Echocardiographic data were analyzed using Vevo Lab 5.7.1. * denotes significantly different from uninfected, *p* < 0.05.

**Figure 3 ijms-26-03801-f003:**
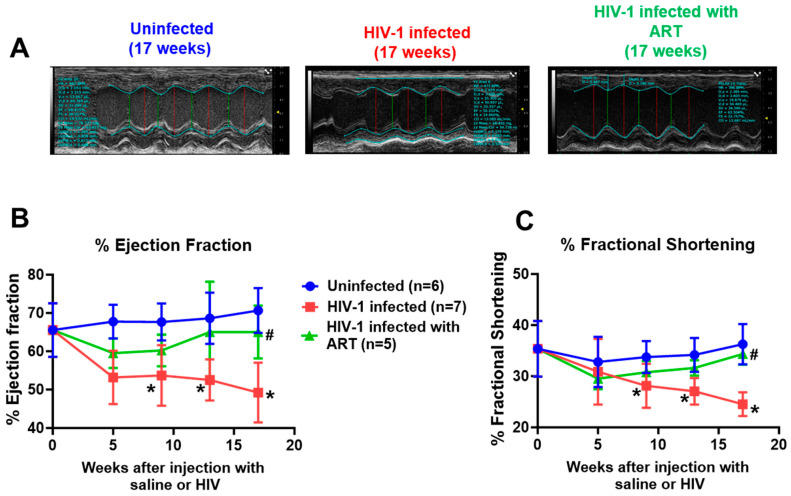
(**A**) shows represented M-Mode echocardiograms from uninfected (n-6), HIV-1-infected (n = 7) and HIV-infected female humanized mice infected with combination DTG/TDF/FTC starting four weeks after productive infection (n = 5). (**B**,**C**) shows longitudinal changes in % ejection fraction and % fractional shortening in uninfected, HIV-1-infected, and HIV-infected female humanized mice infected treated with DTG/TDF/FTC for thirteen weeks, starting four weeks after productive infection. All measurements were taken using Fujifilm Visual Sonics Vevo 3100 system (Fujifilm Visual Sonics, Toronto, ON, Canada) employing a MS550D transducer with a center frequency of 40 Hz and an axial resolution of 40 µM. Echocardiographic data were analyzed using Vevo Lab 5.7.1. * denotes significantly different from uninfected, *p* < 0.05, # denotes significantly different from HIV-1-infected.

**Figure 4 ijms-26-03801-f004:**
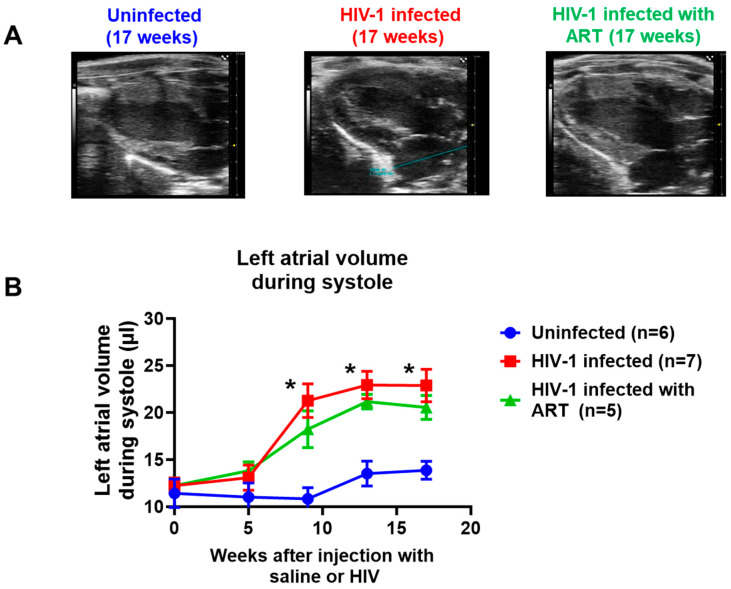
(**A**) shows representative long-axis images of left atria volume in hearts from uninfected (n = 6), HIV-1-infected (n = 7), and HIV-infected female humanized mice infected treated with DTG/TDF/FTC via feed for thirteen weeks, starting four weeks after productive infection (n = 5). (**B**) shows longitudinal changes in left atrial volume in uninfected, HIV-1-infected, and HIV-infected female humanized mice treated with DTG/TDF/FTC for thirteen weeks, starting four weeks after productive infection. All measurements were taken using the Fujifilm Visual Sonics Vevo 3100 system (Fujifilm Visual Sonics, Toronto, ON, Canada) employing a MS550D transducer with a center frequency of 40 Hz and an axial resolution of 40 µM. Echocardiographic data were analyzed using Vevo Lab 5.7.1. * denotes significantly different from uninfected, *p* < 0.05.

**Figure 5 ijms-26-03801-f005:**
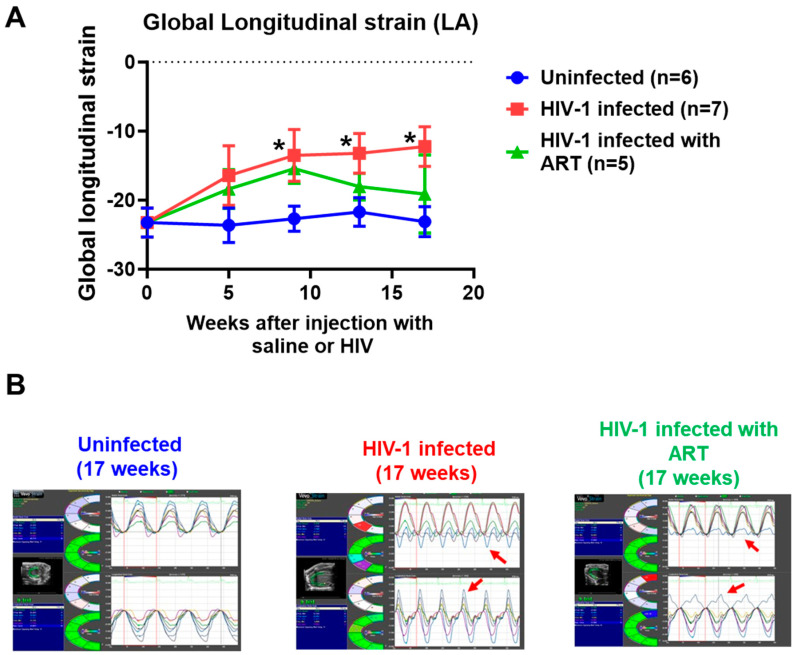
(**A**) shows longitudinal changes in left ventricular strain in uninfected (n = 6), HIV-1-infected (n = 7), and HIV-infected female humanized mice treated with DTG/TDF/FTC, via feed for thirteen weeks, starting four weeks after productive infection (n = 5). (**B**) shows representative strain data in hearts from uninfected, HIV-1-infected, and HIV-infected female humanized mice treated with DTG/TDF/FTC, via feed for thirteen weeks, starting four weeks after productive infection. Red arrows show dyskinesia in opposite segments in hearts from HIV-infected and HIV-infected female humanized mice treated with DTG/TDF/FTC. All measurements were collected using the Fujifilm Visual Sonics Vevo 3100 system (Fujifilm Visual Sonics, Toronto, ON, Canada) employing a MS550D transducer with a center frequency of 40 Hz and an axial resolution of 40 µM. Echocardiographic data were analyzed using Vevo Lab 5.7.1, * denotes significantly different from uninfected, *p* < 0.05.

**Figure 6 ijms-26-03801-f006:**
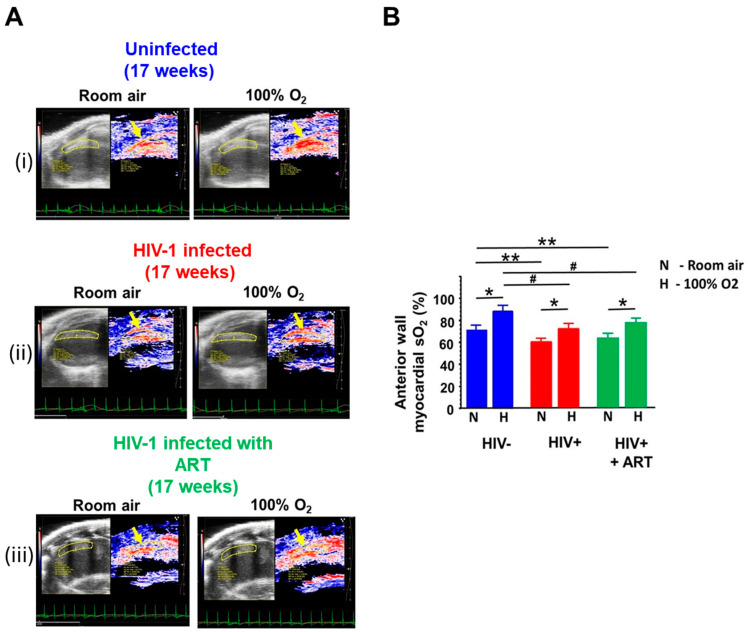
(**A**) shows representative photoacoustic images of saturated oxygenated hemoglobin in the anterior wall of the left ventricle (septum) of hearts from uninfected (n = 6), HIV-1-infected (n = 7) and HIV-infected female humanized mice treated with DTG/TDF/FTC for fourteen weeks (n = 5) in normal air and 100% oxygen (compare yellow arrows). (**B**) shows means ± SEM for n ≥ 5 animals per group. * denotes significantly different between room air and 100% O_2_, *p*  <  0.05, ** denotes significantly different from control in room air, *p*  <  0.05, # denotes significantly different from control with 100% oxygen, *p* < 0.05. All images were collected using the Fujifilm Visual Sonics Vevo 3100 system with a LAZR-X (Fujifilm Visual Sonics, Toronto, ON, Canada), employing a MX550D transducer with a center frequency of 40 Hz and an axial resolution of 40 µM. Photoacoustic image data were analyzed using Vevo Lab 5.7.1.

**Figure 7 ijms-26-03801-f007:**
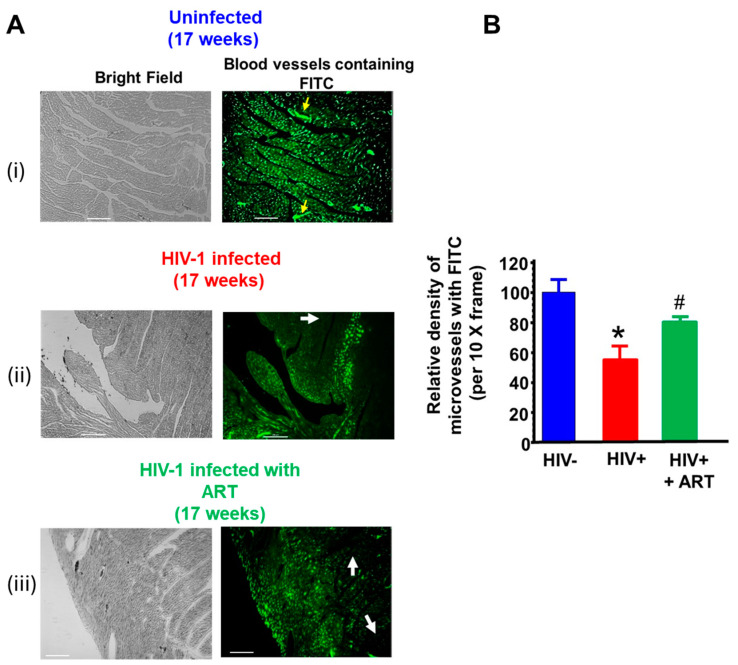
(**A**) shows representative BSA-FITC florescence images of the vasculature in left ventricular wall from uninfected (n = 6), HIV-1-infected (n = 7), and HIV-infected female humanized mice treated with DTG/TDF/FTC for thirteen weeks (n = 5) after four weeks of productive infection. For this, BSA-FITC was injected into a tail vein 5 min prior to euthanasia to visually density of perfused microvessels in heart tissues. Yellow arrows show the BSA-FITC confined within blood vessels. White arrows show regions of ischemia in left ventricular walls of HIV-1-infected and HIV-infected female humanized mice treated with DTG/TDF/FTC. The graph on the right (**B**) shows the relative density of microvessels perfused with BSA-FITC per 10-x frame for n ≥ 5 mice. The white scale bar at the bottom indicates 200 μm. * denotes significantly different from uninfected, *p*  <  0.05, # denotes significantly different from uninfected and HIV-infected, *p* < 0.05.

**Figure 8 ijms-26-03801-f008:**
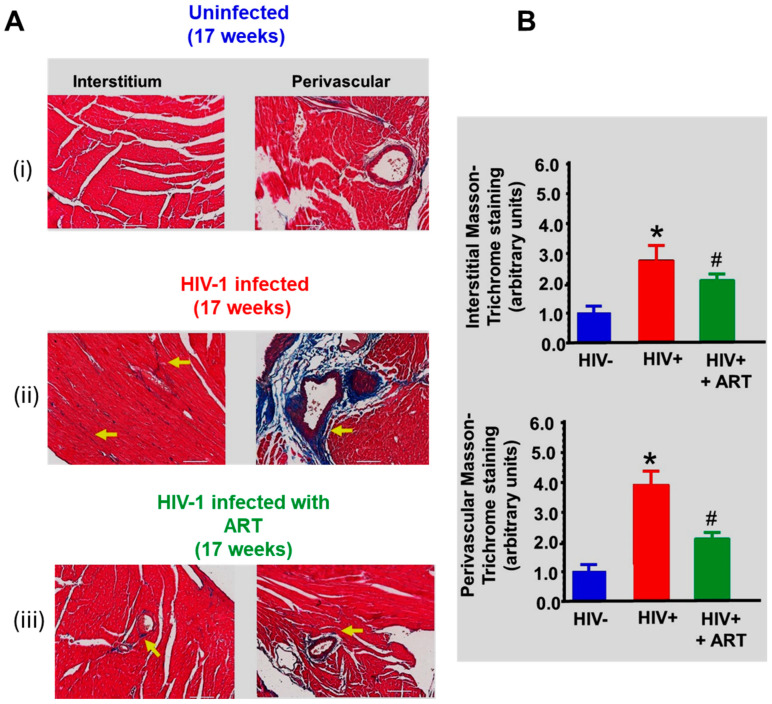
(**A**) shows Masson-Trichrome staining in ventricular walls from uninfected (n = 6), HIV-1-infected (n = 7), and HIV-infected female humanized mice treated with DTG/TDF/FTC for thirteen weeks, starting four weeks after productive infection. Yellow arrows show regions of increased interstitial and perivascular staining in HIV-1-infected and HIV-infected female humanized mice treated with DTG/TDF/FTC. Graphs on the right (**B**) show relative density of Masson-Trichrome staining per 10-x frame for n ≥ 5 animals per group in interstitium and perivascular regions. The white scale bar at the bottom indicates 200 μm. * denotes significantly different from uninfected, *p*  <  0.05, # denotes significantly different from uninfected and HIV-infected, *p* < 0.05.

**Figure 9 ijms-26-03801-f009:**
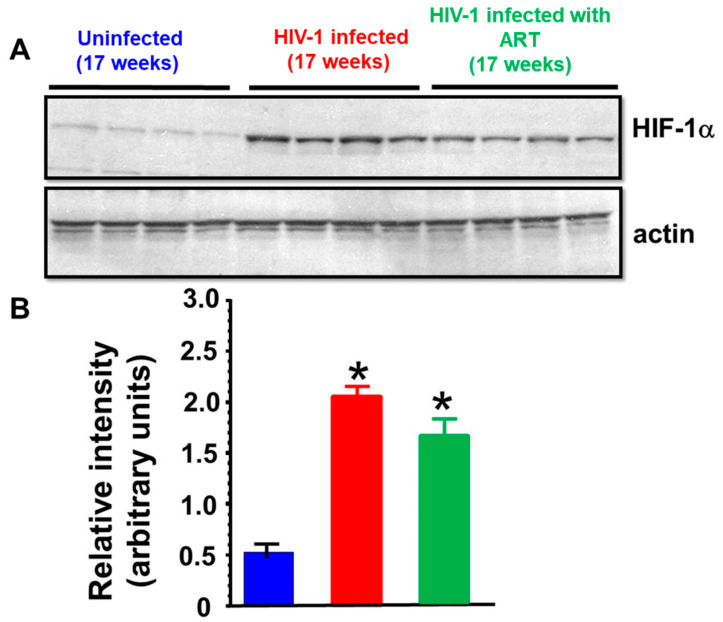
(**A**) shows autoradiograms for HIF-1α and β-actin from ventricular walls from uninfected (n = 4, blue), HIV-1-infected (n = 4, red), and HIV-infected female humanized mice treated with DTG/TDF/FTC for thirteen weeks starting four weeks after productive infection (n = 4, green). The graph in (**B**) shows relative signal intensities of HIF-1α in hearts from uninfected (n = 6), HIV-1-infected (n = 7), and HIV-infected female humanized mice treated with DTG/TDF/FTC for thirteen weeks starting four weeks after productive infection (n = 5). * denotes significantly different from uninfected, *p*  <  0.05.

**Figure 10 ijms-26-03801-f010:**
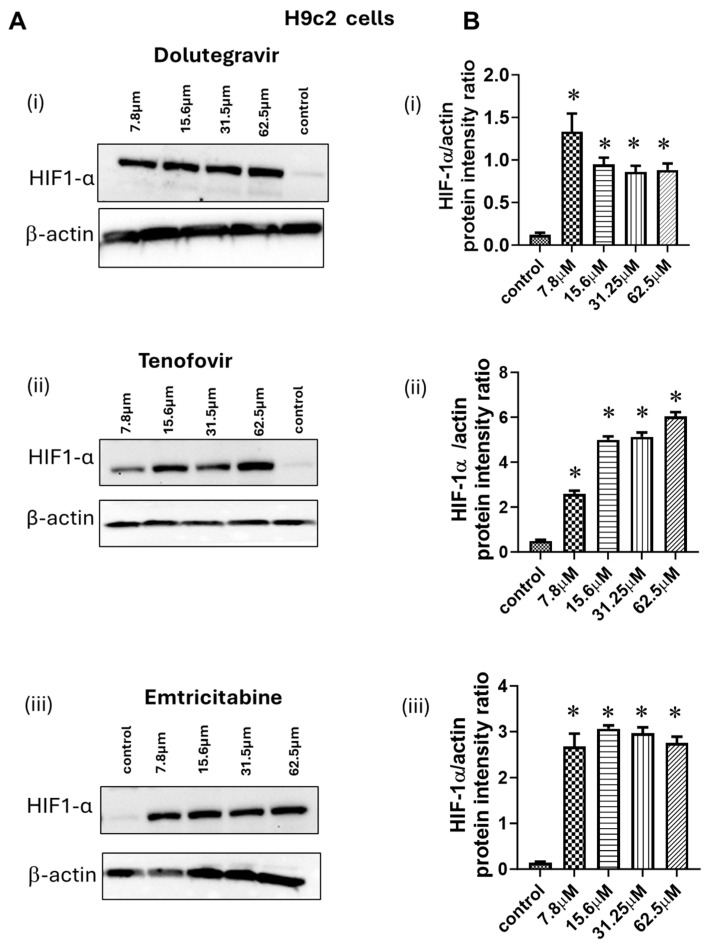
(**A**(**i**–**iii**)) show autoradiograms for total HIF-1α and β-actin from H9C2 cells treated with varying concentrations (7.8 µM to 62.5 µM) of dolutegravir, tenofovir disoproxil fumarate, and emtricitabine for 2 h. at 37 °C. Cells were harvested and Western blots were conducted. The graph in (**B**(**i**–**iii**)) show relative signal intensities of HIF-1α: actin ratios for (n = 3) separate experiments. Data show robust and significant increases in HIF-1α when H9C2 cardiac myocyte cells were exposed to dolutegravir, tenofovir disoproxil fumarate, and emtricitabine. * denotes significantly different from untreated, *p*  <  0.05.

**Table 1 ijms-26-03801-t001:** General characteristics of animals used in the study.

	Parameters	At Start (t = 0 weeks)	End (t = 17 weeks)
Hu-NSG mice (n = 6)	Body weight (g)	18.3 ± 0.4	19.2 ± 1.0
	Plasma HSA-MG (μg/mL)	26.6 ± 3.4	34.1 ± 4.4
	Plasma SSAO activity (units/mL/2 h)	3.1 ± 0.3	3.2 ± 0.4
	Heart weight (mg)	NA	156.2 ± 3.2
Hu-NSG mice infected with HIV-1_ADA_ (n = 6)	Body weight (g)	18.3 ± 0.3	18.9 ± 1.0
	Plasma HIV viral load (RNA copies/mL)	1.0 × 10^5^ (injected)	1.6 ± 0.2 × 10^6^
	Plasma HSA-MG (μg/mL)	25.2 ± 3.2	59.6 ± 5.4 *
	Plasma SSAO activity (units/mL/2 h)	3.1 ± 0.3	6.2 ± 0.2 *
	Heart weight (mg)	NA	143.4 ± 4.2 *
Hu-NSG mice Infected with HIV-1_ADA_ and treated with DTG/TDF/FTC	Body weight (g)	18.3 ± 0.3	19.1 ± 1.0
	Plasma HIV viral load (RNA copies/mL)	1.0 × 10^5^ (injected)	0.5 ± 0.1 × 10^2^
	Plasma HSA-MG (μg/mL)	27.2 ± 2.2	58.1 ± 8.4 *
	Plasma SSAO activity (units/mL/2 h)	3.2 ± 0.1	6.1 ± 0.2 *
	Heart weight (mg)	NA	146.5 ± 4.2 *
	Plasma DTG (ng/mL)	NA	7622.5 ± 665.6
	Plasma TDF (ng/mL)	NA	546.7 ± 74.5
	Plasma FTC (ng/mL)	NA	1818.4 ± 278.4

NA = not applicable, * denotes significantly different from uninfected control (*p* < 0.05).

## Data Availability

Data are available up request.

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
