# Peer review of "Diastolic Dysfunction with Vascular Deficits in HIV-1-Infected Female Humanized Mice Treated with Antiretroviral Drugs"

_ijms, 2025, doi:10.3390/ijms26083801_

Round 1

Reviewer 1 Report

Comments and Suggestions for Authors

-Provide more background on how these findings relate to the broader field of HIV research and cardiovascular health and discuss potential clinical implications in more detail.

-Why did study not included both male and female humanized mice to explore sex differences in the pathophysiology of HIV-related heart disease?

-The study does not discuss the control of environmental factors that may influence heart function, such as diet, physical activity, or stressors on the mice, if no way to add this part now then you need to address it with the imitations part.

-The current study indicates increased HIF-1α expression but lacks detailed investigation into the specific pathways affected by ART that lead to diastolic dysfunction.

-Please take a look for this is recent article published last month may be useful if you add it in your literature.

Innovative Diagnostic Approaches and Challenges in the Management of HIV: Bridging Basic Science and Clinical Practice. https://doi.org/10.3390/life15020209

Author Response

Reviewer #1

Comment #1: Provide more background on how these findings relate to the broader field of HIV research and cardiovascular health and discuss potential clinical implications in more detail.

Response: Thank you for pointing this out.  We have made it clearer in the introduction that rodent models that recapitulate the cardiac pathophysiologies seen in WLWH remain underreported and this has left a void in preclinical studies to evaluate the pharmacologic strategies to alleviate it. Pharmacological strategies to blunt early-onset cardiac diastolic dysfunction in WLWH remain virtually nonexistent at present.

Comment #2: Why did study not included both male and female humanized mice to explore sex differences in the pathophysiology of HIV-related heart disease?

Response: In this study, we were specifically interested in identifying a rodent model that can recapitulate early onset cardiac diastolic dysfunction reported in WLWH. In an earlier study (Sci Rep, 2020 Jun 16;10(1):9746), doi: 10.1038/s41598-020-65943-9), we had reported that female NOD.Cg-PrkdcscidIl2rgtm1Wjl/SzJ humanized mice infected with HIV-1 developed cardiac dysfunction (both diastolic and systolic) earlier than male NOD.Cg-PrkdcscidIl2rgtm1Wjl/SzJ humanized mice infected with HIV-1. This manuscript focused on female mice only as the onset of cardiac diastolic dysfunction starts earlier.  

Comment #3: The study does not discuss the control of environmental factors that may influence heart function, such as diet, physical activity, or stressors on the mice, if no way to add this part now then you need to address it with the imitations part.

Response: Since all three groups of mice were housed in identical conditions and given the same feed composition and enrichment environment, we did not focus on the influence of environmental factors such as diet, physical activity, or stressors on the mice. To this end, we have included a limitation section after the discussion to address these concerns. 

Comment #4: The current study indicates increased HIF-1α expression but lacks detailed investigation into the specific pathways affected by ART that leads to diastolic dysfunction.

Response: We are currently attempting to delineate the mechanisms by which antiretroviral drugs increase the expression of HIF-1α.  However, we have made clearer in the revised manuscript that while the “reduction in density of perfused microvessels is likely to induce HIF-1α, it was not clear if the antiretroviral drugs used, dolutegravir, tenofovir disoproxil fumarate, and emtricitabine were also exacerbating HIF-1α expression.

Comment #5: Please take a look for this is recent article published last month may be useful if you add it in your literature. Innovative Diagnostic Approaches and Challenges in the Management of HIV: Bridging Basic Science and Clinical Practice. https://doi.org/10.3390/life15020209

Response: Thank you We have included this reference in the revised manuscript.

Reviewer 2 Report

Comments and Suggestions for Authors

This manuscript, titled "DIASTOLIC DYSFUNCTION WITH PRESERVED EJECTION FRACTION IN FEMALE HIV-1 INFECTED HUMANIZED MICE TREATED WITH ANTIRETROVIRAL DRUGS", presents multiple issues and limitations that need substantial revision:

The first suggestion is related to the manuscript title. Instead of the given title, the more precise and grammatically sound alternative could be: "DIASTOLIC DYSFUNCTION AND VASCULAR DEFICITS IN HIV-1 INFECTED FEMALE HUMANIZED MICE TREATED WITH ANTIRETROVIRAL THERAPY"

The authors must also reduce the similarity from 39% to below 20%.

The study aims to establish HIV-infected humanized mouse model for diastolic dysfunction, but there is no clear and enough justification for why this specific model is superior to existing ones.

Prior research has already demonstrated cardiovascular complications in HIV patients undergoing antiretroviral therapy, making the novelty questionable.

The manuscript raises concerns about the robustness of the results, making their reliability and validity difficult to assess.

The use of humanized mice is claimed to mimic clinical conditions, yet there is no enough discussion of how well this model replicates human physiology.

The abstract contains multiple grammatical errors, unclear phrasing, and inconsistencies (e.g., "major treat" should be "major threat"; "volunerable" should be "vulnerable").

Terminology is inconsistently used (e.g., “HIF-1a.” is incorrectly punctuated).

The phrase "replicating these clinical features remain under reported" is grammatically incorrect and unclear.

The study concludes that antiretroviral therapeutic drugs contribute to increased HIF-1α expression and diastolic dysfunction but does not provide mechanistic evidence to establish causality.

The manuscript claims that findings mimic conditions in women living with HIV-1, but there is no data comparing the animal model directly with human clinical outcomes.

The conclusion does not clearly state how this study advances current knowledge or its implications for clinical practice.

There is not enough discussion of limitations, potential confounders, or future directions, making the study appear incomplete.

Comments on the Quality of English Language

The English could be improved to more clearly express the research.

Author Response

Reviewer #2

Comment #1: The first suggestion is related to the manuscript title. Instead of the given title, the more precise and grammatically sound alternative could be: "DIASTOLIC DYSFUNCTION AND VASCULAR DEFICITS IN HIV-1 INFECTED FEMALE HUMANIZED MICE TREATED WITH ANTIRETROVIRAL THERAPY"

Response: Thank you. As per suggestion, the title of the revised manuscript is now “DIASTOLIC DYSFUNCTION AND VASCULAR DEFICITS IN HIV-1 INFECTED FEMALE HUMANIZED MICE TREATED WITH ANTIRETROVIRAL THERAPY"

Comment#2: The authors must also reduce the similarity from 39% to below 20%.

Response: The 39% similarity in the original manuscript was due to the methods section. We have edited this section to decrease the similarity.to less than 10%.

Comment #3: The study aims to establish HIV-infected humanized mouse model for diastolic dysfunction, but there is no clear and enough justification for why this specific model is superior to existing ones.

Response:  Thank you for this comment.  HIV-1 infects human cells, and there are a limited number of mouse models capable of being infected with HIV-1 and treated with antiretroviral drugs (see  Nature Reviews Microbiology volume 10, pages852–867 (2012), J AIDS HIV Treat. 2020;2(1):23–29.,   and Mol Neurodegener 2019 Mar 5;14(1):12).  These include Scid-hu–Thy/Liv mice, BLT mice, PBL, NOD scid and mice have been crossed with Il2rg−/− mice (NOD.Cg-PrkdcscidIl2rgtm1Wjl/SzJ), Scid-hu–Thy/Liv mice and BLT mice requires human fetal thymus cells for which we do not have institutional approval. PBL mice have shortened lifespans and are not appropriate for chronic studies as they develop graft versus host disease one month after human cell transplant.

Transgenic rodents that overexpress HIV-1proteins are also available, but these models do not reflect anti-retroviral treatment as HIV proteins are high and they can develop AIDS-Like Disease (J Virol, 1998 Jan;72(1):121–132. doi: 10.1128/jvi.72.1.121-132.1998). Another EcoHIV mice model is also an overexpression model (PNAS  March 8, 2005  vol. 102 no.10  3761).    

Our group uses NOD.Cg-PrkdcscidIl2rgtm1Wjl/SzJ (NSG-humanized mice). These mice were generated by injecting CD34+ hematopoietic stem cells into one-day-old NSG pups, and they generally develop a functional human immune system by 18-20 weeks. At this time, they are adult humanized mice with expression of pan-human CD45, CD3, CD4, CD8, and CD14 cells needed for HIV-1 infection studies. The human immune cells in this model can last for more than one year for chronic HIV studies. This is the first report on cardiac diastolic dysfunction in female HIV-1-infected humanized mice treated with antiretroviral drugs.

Comment #4: Prior research has already demonstrated cardiovascular complications in HIV patients undergoing antiretroviral therapy, making the novelty questionable.

Response: We agree that prior research has already demonstrated cardiovascular complications in HIV patients undergoing antiretroviral therapy. However, it is not permitted to use patients to elucidate basic mechanisms and do drug screening. Hence, the focus of this manuscript is developing/characterizing this in animal (rodent) model, which will have broad clinical application.

Comment #5: The manuscript raises concerns about the robustness of the results, making their reliability and validity difficult to assess.

Response: We respectfully disagree with this comment. The reliability and validity of our data is extremely strong, hence the reason for submitting it for publication. Working with HIV-infected humanized mice is extremely challenging as these mice are immune compromised and a lot of care must be taken to avoid premature death of animals and obtain the data sets in a timely manner by expert personnel using top-class facilities.

Comment #6: The use of humanized mice is claimed to mimic clinical conditions, yet there is no enough discussion of how well this model replicates human physiology.

Response: We appreciate the comment by the reviewer. We have shown in this and in other studies (Sci Rep, 2020 Jun 16;10(1):9746, Front Cardiovasc Med. 2021 Dec 14:8:792180) that non-infected humanized mice exhibit normal cardiac function (E:A ratio, ejection fraction, fractional shortening etc). At present, we do not have grant funding to determine whether our NSG humanized mice replicate human cardiac physiology, and we will do so once funding becomes available. At present, we are funded to investigate cardiac pathophysiologic mechanisms.

Comment #7: The abstract contains multiple grammatical errors, unclear phrasing, and inconsistencies (e.g., "major treat" should be "major threat"; "vulnerable" should be "vulnerable"). Terminology is inconsistently used (e.g., “HIF-1a.” is incorrectly punctuated).The phrase "replicating these clinical features remain under reported" is grammatically incorrect and unclear.

Response: Thank you, we have corrected them in the revised manuscript.

Comment #8: The study concludes that antiretroviral therapeutic drugs contribute to increased HIF-1α expression and diastolic dysfunction but does not provide mechanistic evidence to establish causality.

Response: Thank you. We are currently working to elucidate the mechanisms by which antiretroviral drugs increase HIF-1a. We have made it clearer in the revised manuscript that the micro-ischemia could account for increased expression of HIF-1a.  However, how much the antiretroviral drugs are contributing to it needs further investigation.

Comment #9: The manuscript claims that findings mimic conditions in women living with HIV-1, but there is no data comparing the animal model directly with human clinical outcomes.

Response: Thank you. This is a limitation of the study as we do not have autopsied human cardiac tissues to validate the fibrosis and HIF-1a.  We are currently in the process of obtaining autopsied human cardiac tissues from uninfected and HIV-infected individuals.

Comment #10: The conclusion does not clearly state how this study advances current knowledge or its implications for clinical practice.

Response: We appreciate this comment. Identifying a model that could recapitulate the key features of cardiac dysfunction in WLWH, including left ventricular diastolic dysfunction, vascular deficits, myocardial infarction, and fibrosis, could help to identify mechanisms that contribute to heart failure and therapeutic strategies to blunt its development.  To the best of our knowledge, the underlying causes for early onset heart failure in WLWH remain poorly understood.

Comment #11:  There is not enough discussion of limitations, potential confounders, or future directions, making the study appear incomplete.

Response: We have included a section on limitations and future directions in the revised manuscript. Thank you

Round 2

Reviewer 2 Report

Comments and Suggestions for Authors

There are still some specific comments that need to be addressed.

1) Humanized mice are claimed to mimic clinical conditions, yet there is not enough discussion of how well this model replicates human physiology.

2) The study concludes that antiretroviral therapeutic drugs contribute to increased HIF-1α expression and diastolic dysfunction but does not provide mechanistic evidence to establish causality.

3) The manuscript claims that findings mimic conditions in women living with HIV-1, but no data compares the animal model directly with human clinical outcomes.

Author Response

The authors thank Reviewer #2 for his insightful comments that have helped us make clearer the findings of the study.

Comment #1: Humanized mice are claimed to mimic clinical conditions, yet there is not enough discussion of how well this model replicates human physiology.

Response: We have removed the one-time use of the word “replicate” from the abstract and the one-time use of the word “mimic” from the authors contribution section and made clearer in the R2 version that the diastolic dysfunction (DD), vascular deficits, myocardial infarction (ischemia) and fibrosis reported in WLWH are also seen in HIV-infected NSG humanized mice treated with DTG/FTC/TDF for 13 weeks. What remains to be determined are the mechanisms responsible for vascular deficits, myocardial infarction (ischemia) and fibrosis in this model.  They may or may not be the same as in other models and in humans.  We have also provided below a thorough explanation of the advantages and disadvantages of using rodent models to study human diseases.  In the R2 manuscript the limitation section has been expanded since no mouse model cannot exhibit 100% of human physiology/pathophysiology.

Rodent models (mouse/rat) are widely used in preclinical biomedical research to elucidate mechanisms that may be contributing to the human diseases. They are relatively inexpensive and can be manipulated genetically and pharmacologically to characterize disease pathogenesis and treatment. To date, no single rodent model can 100% replicate all pathophysiological conditions reported in human.  However, select rodent models have been used to study mechanisms that drive specific human diseases.  One of these diseases is HIV-associated heart failure.  While mouse hearts beat 500 X per minute and human hearts beat ~80 X per minute, mechanisms that drive automaticity, contraction and relaxation are >90% similar. Circulation: Cardiovascular 2011, 4(1): 2-4 and Pediatr Res. 2014 Dec;76(6):500-7)

HIV-1 is a human specific virus that infects cells of the immune system that contain the CD4 protein, including CD4+-T cells, monocytes, macrophages, dendritic cells, and microglia.  HIV-1 does not infect mouse immune cells. Some groups have developed transgenic rodent models that overexpress HIV-1 proteins and use these models to study their impact on cardiac function  (J Virol, 1998 Jan;72(1):121–132., PNAS  March 8, 2005  vol. 102 no.10  3761).

Our group uses the NOD.Cg-PrkdcscidIl2rgtm1Wjl/SzJ (NSG) humanized mouse model since these mice can be infected with HIV-1 and treated with antiretroviral drugs to lower HIV-1 viremia. One day after birth, NSG mice are irradiated to destroy their immune cells, and hepatically injected with human CD34+ hematopoietic stem cells. Twenty weeks later irradiated mice develop a functional human immune system, and express pan-human CD45, CD3, CD4, CD8, and CD14 cells that  can be infected with HIV-1. During the humanization process, <5% of animals develop graft vs host diseases, and these mice were not used in our experimental process. Cardiac contraction and relaxation kinetics in NSG humanized mice are similar to that of NSG mice in terms of E:A ratio (~1.2-1.3), ejection fraction (65-70%), fractional shortening (35%).

Following HIV-1 infection, cardiac function of NSG humanized mice decline over time, mitral regurgitation, increase in E:A ratio, reductions in ejection fraction and fractional shortening and enlarged left atria. These changes have also been reported in PLWH.  Non-infected humanized NSG mice of similar ages do not exhibit cardiac dysfunction, indicating that changes arise from HIV-1 infection.

This mouse model is just a model and is not expected to exhibit all aspects of human cardiac physiology and  pathophysiology.  What it does exhibit after HIV-1 infection and treatment with ARDs, is the cardiac diastolic dysfunction, impaired microvascular perfusion/ischemia, and fibrosis reported in WLWH. There are several limitations with NSG humanized mice (i) B cells and lymph nodes are not fully developed, (ii) drug metabolism may be different in humanized mice and humans due to differences in liver enzyme expression, which could affect drug clearance and toxicity profiles and (iii) lifespan is shorter and accelerated which can affect disease progression and therapeutic response studies.

We have made clearer that the diastolic dysfunction with vascular deficits, myocardial infarction, and fibrosis reported by several investigators in WLWH can be seen in our HIV-infected humanized treated with ARD.  Whether the cause(s) for the changes in mouse are the same as in human remains to be determined.   

Comment #2: The study concludes that antiretroviral therapeutic drugs contribute to increased HIF-1α expression and diastolic dysfunction but does not provide mechanistic evidence to establish causality.

Response: We did not conclude in the R1 manuscript that “the antiretroviral therapeutic drugs contribute to diastolic dysfunction” and this point is made clearer in the R2 manuscript.  What we did say in the R1 manuscript is that in HIV-1 infected humanized mice treated with anti-retroviral drugs developed diastolic dysfunction.  We also said that cardiac tissues from HIV-1 infected humanized mice treated with dolutegravir/tenofovir disoproxil fumarate/emtricitabine exhibited a reduction in the density of perfused microvessels and increased levels of HIF-1a. We also found that treating H9C2 cells with  dolutegravir, tenofovir disoproxil fumarate and emtricitabine lead to an increase in HIF-1a.  Whether there is a mechanistic link between increased HIF-1α expression and diastolic dysfunction remains to be determine. 

With regards to specific mechanisms by which dolutegravir (an integrase strand transfer inhibitor), tenofovir disoproxil fumarate (a nucleotide reverse-transcriptase inhibitor) and emtricitabine (a nucleoside reverse transcriptase inhibitor) increase HIF-1a in H9C2 cells. What we know from the literature is that HIF-1a is continuously synthesized and transported to the cytoplasm of cells where it gets hydroxylated by prolyl hydroxylase and target for proteasomal degradation (J Physiol 2021 Jan;599(1):23-37). Reactive oxygen species (ROS) is a potent inhibitor of prolyl hydroxylase.  Dolutegravir, tenofovir disoproxil fumarate and emtricitabine are potent inducers of ROS production and the increase in HIF-1a seen in H9C2 cells may be due in part to increases in ROS by dolutegravir, tenofovir disoproxil fumarate and emtricitabine (Molecules, 2022 Dec 19;27(24):905 and Life Sci 2022 Apr 1:294:120329).  Whether dolutegravir, tenofovir disoproxil fumarate and emtricitabine are directly binding to inhibiting prolyl hydroxylase remain to be investigated. We have made clearer that the increase in HIF-1a is total HIF-1a.  

Comment #3: The manuscript claims that findings mimic conditions in women living with HIV-1, but no data compares the animal model directly with human clinical outcomes.

Response: In the abstract, of the R2, we remove the word recapitulate.  It now reads “These data show that HIV-infected Hu-mice treated with DTG/TDF/FTC for thirteen weeks, develop cardiac diastolic dysfunction, with vascular deficits, ischemia and fibrosis like that reported in women living with HIV-1 infection (WLWH). Cardiac diastolic dysfunction, with vascular deficits, ischemia and fibrosis have been published by several groups including CirculationVolume 147, Issue 1, 3 January 2023; Pages 83-100, Front Cardiovasc Med. 2025 Jan 29:12:1534533, Curr Opin HIV AIDS. 2022 Jul 5;17(5):270–278).  

Thank you

Round 3

Reviewer 2 Report

Comments and Suggestions for Authors

All my suggestions and comments have been addressed.